**Data Availability Statement:** All relevant data are within the manuscript and its Supporting Information files.

**Funding:** This work was supported by the NIH R03 (DK103964) grant, R21 (AI137663) grant, and Eli

# High circulating elafin levels are associated with Crohn's disease-associated intestinal strictures

Jiani Wang[1,2], Christina Ortiz[1], Lindsey Fontenot[1], Ying Xie[1,2], Wendy Ho[1], S. Anjani Mattai[3], David Q. Shih[4], Hon Wai Koon [1]*

**1** Vatche and Tamar Manoukian Division of Digestive Diseases, David Geffen School of Medicine at the University of California Los Angeles, Los Angeles, CA, United States of America, **2** Department of Gastroenterology, First Affiliated Hospital, China Medical University, Shenyang City, Liaoning Province, China, **3** Department of Medicine, David Geffen School of Medicine at the University of California Los Angeles, Los Angeles, CA, United States of America, **4** F. Widjaja Foundation, Inflammatory Bowel & Immunobiology Research Institute, Cedars-Sinai Medical Center, Los Angeles, CA, United States of America

* hkoon@mednet.ucla.edu

## Abstract

### Background

Antimicrobial peptide expression is associated with disease activity in inflammatory bowel disease (IBD) patients. IBD patients have abnormal expression of elafin, a human elastase-specific protease inhibitor and antimicrobial peptide. We determined elafin expression in blood, intestine, and mesenteric fat of IBD and non-IBD patients.

### Methods

Serum samples from normal and IBD patients were collected from two UCLA cohorts. Surgical resection samples of human colonic and mesenteric fat tissues from IBD and non-IBD (colon cancer) patients were collected from Cedars-Sinai Medical Center.

### Results

High serum elafin levels were associated with a significantly elevated risk of intestinal stricture in Crohn's disease (CD) patients. Microsoft Azure Machine learning algorithm using serum elafin levels and clinical data identified stricturing CD patients with high accuracy. Serum elafin levels had weak positive correlations with clinical disease activity (Partial Mayo Score and Harvey Bradshaw Index), but not endoscopic disease activity (Mayo Endoscopic Subscore and Simple Endoscopic Index for CD) in IBD patients. Ulcerative colitis (UC) patients had high serum elafin levels. Colonic elafin mRNA and protein expression were not associated with clinical disease activity and histological injury in IBD patients, but stricturing CD patients had lower colonic elafin expression than non-stricturing CD patients. Mesenteric fat in stricturing CD patients had significantly increased elafin mRNA and protein expression, which may contribute to high circulating elafin levels. Human mesenteric fat adipocytes secrete elafin protein.

and Edythe Broad Foundation to Hon Wai Koon. The funder was not involved in the design of the study and collection, analysis, and interpretation of data and in writing the manuscript.

**Competing interests:** The authors have declared that no competing interests exist.

## Conclusions

High circulating elafin levels are associated with the presence of stricture in CD patients. Serum elafin levels may help identify intestinal strictures in CD patients.

## Introduction

Intestinal stricture formation is a debilitating complication of inflammatory bowel disease (IBD) [1]. Chronic inflammation in Crohn's disease (CD) patients leads to multiple cycles of tissue injury and healing [2]. The transforming growth factor-beta 1 (TGF-β1)-activated myo-fibroblasts produce an excessive amount of extracellular matrix, such as collagen, in the sub-mucosa and mucosa [3], which obstructs bowel movement.

Around one-third of CD patients develop strictures (Vienna classification B2) over ten years after diagnosis [4]. The IBDchip European project that included 1528 CD patients with more than ten years of follow-up showed 48.2% of patients with stricturing behavior [5]. Imaging and endoscopic evaluations of intestinal strictures are expensive and time-consuming [6, 7]. Several serum factors (miR-19, miR29, collagen, fibronectin, tissue inhibitor of matrix metalloproteinase-1, basic fibroblasts growth factor, chitinase 3-like 1 (YKL-40), anti-*Saccharomyces cerevisiae* antibodies, and fibrocytes) had shown conflicting results with low specificity for stricturing CD patients [8]. We are interested in discovering novel biomarkers for intestinal strictures because there are none established for indicating the presence of intestinal strictures.

Antimicrobial peptides and proteins such as serum cathelicidin, stool lactoferrin, and fecal calprotectin (FC) demonstrated clinical utilities as IBD biomarkers [9, 10]. Fecal calprotectin is useful for assessment of IBD disease activity [11]. Cathelicidin has anti-inflammatory and anti-fibrogenic effects in colitis models [12–15]. Elafin is a small (6kDa) elastase-specific protease inhibitor with antimicrobial functions, primarily expressed in immune cells, intestinal tract, vagina, lungs, and skin [16]. Increased serum elafin levels are significantly associated with the presence of rheumatoid arthritis and the diseased area of psoriasis [17, 18].

Colonic elafin mRNA expression was increased in ulcerative colitis (UC) patients [19]. However, there was no increase of colonic elafin mRNA and protein expression in CD patients [20]. Zhang's group reported the reduced elafin mRNA expression in peripheral blood leukocytes of IBD patients [21, 22]. However, the relevance of elafin in intestinal strictures is unknown. Interestingly, UVA irradiation induces elafin expression in dermal fibroblasts, leading to the accumulation of elastic fibers in the actinic elastosis of sun-damaged skin [23]. Therefore, this evidence suggests the potential association between elafin and fibrogenesis.

A recent study suggests that mesenteric fat wrapping (creeping fat) may be associated with the risk of intestinal stricture in CD patients, but the mechanism of this association is unknown and has not yet been identified [24]. Elafin expression in the adipose tissue of IBD patients is unknown. We hypothesize that a link between elafin expression and intestinal fibrosis may exist. This study examined the expression of elafin in circulation, intestine, and mesenteric fat in non-IBD, UC, stricturing CD, and non-stricturing CD patients.

## Materials and methods

### Human serum samples

For serum samples, IBD patients of cohort 1 were recruited from UCLA Gastroenterology clinic, and control normal patients of cohort 1 were recruited from UCLA Internal Medicine

clinic. This cohort consists of 50 healthy, 23 UC, and 28 CD patients (S1 Table). All serum samples of cohort 1 were prepared by UCLA Department of Pathology. All serum samples from cohort 2 were obtained from UCLA Center for IBD Biobank, which consists of 20 healthy, 57 UC, and 67 CD patients. Patients of these two cohorts did not overlap. All samples were collected during the indicated diagnostic procedure between 2012–2015 prospectively. The serum sample study was approved by the UCLA Institutional Review Board (protocol number IRB 12–001499 and IRB 13–001069). Written informed consent was obtained from all subjects by either UCLA Pathology or IBD Center. Separate informed consent was waived by UCLA IRB.

Inclusion criteria: IBD diagnosis was confirmed by UCLA gastroenterologists. Both cohorts included patients with a wide range (from remission to severe) of clinical and endoscopic disease activity. Intestinal strictures in CD patients were identified by magnetic resonance enterography (MRE), computed tomography (CT), or endoscopy. The intestinal strictures in CD patients were defined by prestenotic dilation, luminal narrowing, and increased wall thickness. The gastroenterologists requested the IBD patients to blood collection procedures as medically indicated. The internal medicine physician requested the normal patients to blood collection procedures as medically indicated. The healthy subjects (control group) visited the UCLA Internal Medicine clinic for regular body checkups. The healthy subjects did not have concurrent cancer, infection, obesity (BMI>30), prediabetes, or diabetes.

Exclusion criteria: Pregnant women, prisoners, or minors under age 18 were not included. Additionally, IBD patients with concurrent acute infection (CMV, *C. difficile*, and tuberculosis) and malignant conditions were excluded. Serum samples with hemolysis were excluded.

## Human colonic and mesenteric fat samples

Patient-matched human colonic and mesenteric fat samples were collected from the Cedars-Sinai Medical Center [25]. This cohort consists of 40 non-IBD, 52 UC, 28 non-stricturing CD, and 15 stricturing CD patients (S3 Table). All colonic and mesenteric fat samples were collected at the same time from surgical operations. All samples were collected during the indicated diagnostic procedure between 2010–2014 prospectively. The colonic and fat sample study was approved by institutional review boards (Cedars-Sinai Institutional Review Board, IRBs 3358 and 23705, and UCLA Institutional Review Board IRB-11-001527). Written informed consent was obtained from all subjects by the Cedars-Sinai Medical Center. Separate informed consent was waived by UCLA IRB. All methods were carried out in accordance with relevant guidelines and regulations.

Inclusion criteria: IBD diagnosis was confirmed by Cedars-Sinai Medical Center gastroenterologists. The Cedars-Sinai Medical Center gastroenterologists referred the patients to surgical procedures, as medically indicated. These IBD patients mostly had severe disease activity or severe strictures after drug treatments that were justified for surgical resection. Colonic and mesenteric fat samples of IBD patients were collected during surgical removal of diseased tissues. Full-thickness involved regions of colonic tissues were used in this study. Colonic and mesenteric fat samples of control group patients were collected during surgical removal of colonic tumors and adjacent normal tissues. The colonic and mesenteric fat with normal histological structures were used as non-IBD control tissue samples. The presence of strictures in the colonic tissue was confirmed by the Cedars-Sinai Medical Center pathologists.

Exclusion criteria: Pregnant women, prisoners, or minors under age 18 were not included. Additionally, IBD patients with concurrent acute infection (CMV, *C. difficile*, and tuberculosis) and malignant conditions were excluded. Colonic and mesenteric fat samples of bad tissue quality or without significant proportions of mucosa were not included.

## Serum exosome preparation

Serum exosomes were prepared by total exosome isolation reagent (#4478360, ThermoFisher) and quantified by bicinchoninic acid (BCA) protein assay (#23225, ThermoFisher).

## ELISA

Human colonic tissues were homogenized in RIPA buffer with a protease inhibitors cocktail (sc-24948, Santa Cruz Biotechnology). Human sera were diluted ten-fold with reagent diluent and added to the ELISA plates. Elafin was detected with an ELISA kit (DY1747 R&D Systems) as described previously [26]. Serum cytokines were detected with multiplex ELISA (human 27-plex #m500kcaf0y, Bio-Rad).

## Whole transcriptome RNA sequencing of human colonic RNA samples

Colonic total RNA samples from two stricturing CD and two non-stricturing CD patients (Cedars-Sinai Medical Center) were used for next-generation whole-transcriptome RNA sequencing (Omega Bioservices). Library was prepared by Illumina TruSeq Stranded mRNA library prep. Sequencing was run on HiSeq 4000/x Ten platform in PE 2x150 format with 5 million reads per sample.

## Cell cultures

Human CCD-18Co intestinal fibroblasts (ATCC) (2 x $10^6$ cells/plate) were cultured in minimal essential medium Eagle's medium (MEM) containing 10% fetal bovine serum (FBS) and 1% penicillin-streptomycin (P/S) (Invitrogen) [15, 25]. Serum-starved CCD-18Co cells were treated with 15μg/ml of anti-elafin neutralizing antibody (AF1747, R&D Systems) or control antibody (AB-108-C, R&D Systems), followed by exposure to human sera from normal, UC, stricturing CD, and non-stricturing CD patients (100μl/mL). CCD18Co fibroblasts in MEM were incubated with 100μg/ml of human serum exosomes for 24 hours. Human serum exosomes were obtained from 12 patients per group. For inhibition of miR-205-5p, serum-starved fibroblasts were pretreated with either 50nM control (YI00199006) or miR205-5p (YI04101508-DDA) power inhibitors (Qiagen) for 24 hours. Power inhibitors were dissolved in Tris-EDTA buffer. The final concentration of miRNA inhibitor in cell culture was 50μM.

Human primary intestinal fibroblasts (two CD patients) were isolated, as described previously [27]. The primary fibroblasts (1 x $10^6$ cells/plate) were cultured in Dulbecco's modified Eagle media (DMEM) containing 10% fetal bovine serum and 1% penicillin/streptomycin and serum-starved overnight before experiments [15].

Peripheral blood mononuclear cells (PBMCs) were obtained from a healthy donor (C-12907, Promocell). PBMCs in mononuclear cell medium (C-28030, Promocell) were incubated with 100μg/mL of human serum exosomes for 24 hours. Human serum exosomes were obtained from 6 patients per group. At the end of the experiments, the treated PBMCs were centrifuged, and the cell pellets were used for RNA extraction.

## Human mesenteric fat adipocyte experiments

Mesenteric fat preadipocytes from non-IBD, CD, and UC patients were collected from a previous study and stored in liquid nitrogen [28]. The human preadipocytes were thawed and cultured in DMEM/F12 media containing 10% calf serum and 1% P/S (Invitrogen) until >60% confluence was achieved. The preadipocytes were dissociated by trypsin/EDTA solution (Invitrogen) and seeded to 6-well plates (400,000 cells per plate) in DMEM/F12 media containing 10% calf serum and 1% P/S. Two days later, the preadipocytes underwent differentiation

process by incubating with induction media (DMEM with FBS, P/S/G, bovine insulin (Sigma I-5500; 1μg/mL), dexamethasone (Sigma D-4902; 1μM) and isobutylmethylxanthine (IBMX; Sigma I-5500; 115μg/mL) for two days, insulin media (DMEM with FBS, P/S/G and insulin (1 μg/mL)) for two days, and DMEM + FBS + P/S for two days [29]. The adipocytes were regarded as differentiated by the observation of lipid droplet deposition under microscope. The differentiated adipocytes were serum-starved for 8 hours, followed by incubation with human serum exosomes (100μg/mL) for 16 hours. The conditioned cells were then switched to serum-free DMEM media for 6 hours to let the cells secrete elafin. The conditioned media were collected for elafin ELISA.

## Histological evaluation of intestinal injury

We prepared paraffin-embedded sections of each human colonic biopsies at UCLA Tissue Processing Core Laboratory (TPCL). Paraffin-embedded sections were cut at 4μm thickness, and H&E staining of the tissue sections was performed as stated before [15, 25]. Microphotographs were recorded at multiple locations and blindly scored by two investigators using a previously described scoring system [30].

## Elafin immunohistochemistry

Elafin immunohistochemistry of human colonic and mesenteric fat tissues was performed by TPCL. Paraffin was removed with xylene. The sections were then rehydrated through graded ethanol. Endogenous peroxidase activity was blocked with 3% hydrogen peroxide in methanol for 10 minutes. Heat-induced antigen retrieval (HIER) was carried out for all sections in 0.01M citrate buffer, pH = 6 using a Biocare decloaker at 95˚C for 25 minutes. After treatment with blocking buffer (2% BSA) for 1 hour, the slides were then incubated overnight at 4˚C with rabbit polyclonal to elafin in 2% BSA at 1:100 dilution (Sigma, HPA017737). The signal was detected using the rabbit horseradish peroxidase EnVision kit (DAKOCytomation, K4003). This secondary antibody kit was directly applied to the slides without dilution. All sections were visualized with the diaminobenzidine reaction and counterstained with hematoxylin. Images were taken with a Zeiss AX10 microscope in a blind manner.

## Quantitative real-time RT-PCR

Total RNA was isolated by an RNeasy kit (#74104, Qiagen) and reverse transcribed into cDNA by a high-capacity cDNA RT kit (#4368813, ThermoFisher). Quantitative PCR reactions were run with Fast Universal PCR master mix (#4352042, ThermoFisher) in a Bio-Rad CFX384 system [26]. The mRNA expression was determined by using cataloged primers (ThermoFisher) for human collagen COL1A2 (Hs01028956_m1), alpha smooth muscle actin ACTA2 (Hs00426835_g1), transforming growth factor-beta one TGF-b1 (Hs00998133_m1), and elafin PI3 (Hs00160066_m1). Relative mRNA quantification was performed by comparing test groups and normal control group, after normalization with endogenous control gene human 18S (Hs99999901_s1).

Preliminary screening for the presence of serum exosomal miRNAs was determined using a miScript human miFinder PCR array (MIHS-001Z, Qiagen). RNA was converted to cDNA using miScript RT kit (218060). PCR reactions were performed with miScript SYBR Green PCR kit (218073). Since many miRNAs in the PCR arrays were undetectable in serum exosomes, we selected the detectable miRNAs and determined their relative expression using miRCURY LNA miRNA PCR assays. RNA was converted to cDNA using miRCURY LNA RT kit (339340) and PCR reactions were run with miRCURY LNA SYBR Green PCR kit (339346). The miRNA expression was detected using Qiagen miRCURY PCR assays. Relative miRNA

quantification was performed by comparing test groups and normal control group, after normalization with housekeeping miRNA (RNU1A1). The measurement of miRNA was determined by miRCURY LNA PCR assays. All miRNA-related reagents were purchased from Qiagen.

The fold changes are expressed as $2^{\Delta\Delta Ct}$. Fold-change values greater than one indicate a positive- or an up-regulation, and the fold-regulation is equal to the fold-change. Fold-change values less than one indicate a negative or down-regulation, and the fold-regulation is the negative inverse of the fold-change.

## Power analysis

Serum sample study: At least 30 patients per group were required to achieve a statistically significant difference of serum elafin levels between control (7939pg/ml), UC (12987pg/ml), and CD (12344pg/ml) patients with standard deviation = 4860, alpha = 0.5, and power = 0.8. The combined dataset from the two serum cohorts satisfied this requirement.

Surgical sample study: At least 30 patients per group were required to achieve a statistically significant difference of colonic elafin mRNA expression between control (1.39 fold), UC (12 fold), and CD (4 fold) patients with standard deviation = 2.55, alpha = 0.5, and power = 0.8. Our cohort satisfied this requirement.

## Statistical analysis

Colonic elafin mRNA and protein expression were arranged in low-to-high order. The entire range of data was divided into three equal tertiles (⅓, ⅓, ⅓). Serum elafin concentrations were arranged in order from low to high. We compared the performance of multiple cut-off points of elafin levels at each disease parameter for optimization of test performance. After many calculations using various cutoff points, the optimized universal cut-off points yielding the highest area under the curve (AUC) values in receiver operating characteristic (ROC) curves (most accurate) were shown in this study. Calculation of prevalence of the disease, sensitivity, specificity, positive predictive value, negative predictive value, and relative risk was described previously [10]. AUCs of ROC curves were calculated online (easyROC web-tool, www.jrocfir.org, and Microsoft Azure Machine Learning Studio). Unpaired Student's *t*-tests were used for two-group comparisons of continuous data (GraphPad QuickCalcs) online. One-way ANOVAs with Tukey Honestly Significant Difference *post-hoc* tests were used for multiple-group comparisons (Statpages) online. Bar graphs and scatter plots were made using Microsoft Excel. Results were expressed as mean +/- SEM. Significant *p* values are shown in each figure.

## Machine learning algorithm for indicating the presence of stricturing in CD patients

The combined CD cohort dataset containing 67 CD patients in CSV file format was loaded into the Microsoft Azure Machine Learning Studio. The dataset included serum elafin level and 14 clinical parameters, i.e., patient's age at blood collection (number), years of disease duration (number), C-reactive protein/CRP (number), ESR (number), HBI (number), count of IBD-related surgery (number), gender (male or female), smoking habit (yes or no), use of biologics (yes or no), use of steroid (yes or no), use of immunomodulator (yes or no), use of aminosalicylate (yes or no), presence of fistula (yes or no), and presence of stricture (yes or no).

The machine learning algorithm included the clinical data based on their relevance for the accurate indication of intestinal strictures. The entire dataset was split into 50% for training and 50% for evaluation. The trained model was built on a two-class decision forest algorithm.

The algorithm utilized default parameters including bagging resampling method, single parameter create trainer mode, 8 decision trees, 32 maximum depth of the decision trees, 128 random splits per node, and 1 minimum number of samples per leaf node. The scored dataset showed score probability (0–0.5 indicates no stricture, 0.51–1.0 indicates stricture), scored labels (yes or no stricture), and AUC values of ROCs.

## Results

### High serum elafin levels indicated an elevated risk of stricture in CD patients

Baseline characteristics, disease locations, and medication use of the two serum sample cohorts are shown in S1 Table. Demographic profiles, disease conditions, and medication uses of these two cohorts are comparable, but not the same. All IBD patients were not treatment naïve, but 20% of UC patients and 17% of CD patients did not have concurrent medication at the time of blood collection. Medication use statistics are shown in S2 Table. Our study (80 UC and 95 CD) included more IBD patients than several other antimicrobial peptide-IBD studies [21, 31, 32].

The detected serum elafin levels in nanogram per milliliter range were similar to the findings of other elafin-related studies [17, 33]. UC patients in both cohorts had significantly higher serum elafin levels than control patients (Fig 1A). There was a trend of mildly increased serum elafin levels in CD patients, but the difference was not statistically significant (Fig 1A). We combined the datasets of both cohorts to yield a sample size for statistical analysis. The combined dataset had also been used in our previous cathelicidin-IBD biomarker study [10]. Serum elafin levels were directly proportional to the Harvey Bradshaw Index (HBI) in CD patients (Fig 1B), but linear regression suggests a weak positive correlation (low $R^2$ value). Serum elafin levels were not associated with the Simple Endoscopic Score for Crohn's Disease (SES-CD) (Fig 1C).

Cohort 1 had 50% stricturing CD patients, while cohort 2 had 24% stricturing CD patients (S1 Table). The number of stricturing CD patients in individual cohorts was below the required sample size to achieve statistical significance. Separate calculations of individual cohorts showed increased serum elafin levels in stricturing CD patients, but the differences were statistically insignificant (S1A Fig). The combined dataset shows that stricturing CD patients had significantly higher serum elafin levels than non-stricturing CD patients (Fig 1D). The high elafin group had a significantly higher relative risk (RR = 2.45) than the low elafin group in having intestinal strictures at the time of blood collection (Fig 1E). However, the serum elafin levels in CD patients with and without fistulas were similar, suggesting that serum elafin levels are not associated with the occurrence of fistulas in CD patients (S1B Fig).

We found no association between elafin levels, age (A1-3), disease location (L1-4), use of medication, and body mass index (BMI) at the time of blood collection (S2 Table, panel A-E). However, stricturing CD patients had significantly longer duration of disease than non-stricturing CD patients did (S2 Table, panel F). Among stricturing CD patients, the high serum elafin group also had significantly longer duration of disease than the low serum elafin group (S2 Table, panel F).

### Machine learning algorithm improves the accuracy of elafin for indicating strictures in CD patients

To evaluate whether circulating elafin alone is a good indicator for intestinal stricture, we determined its accuracy with ROC analysis. Serum elafin alone is moderately accurate for

A

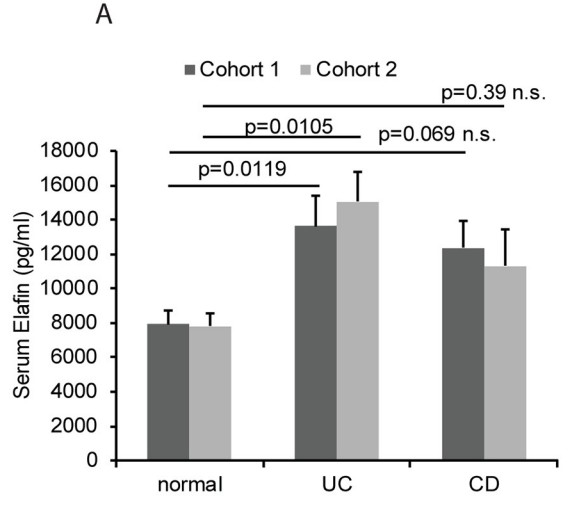

B

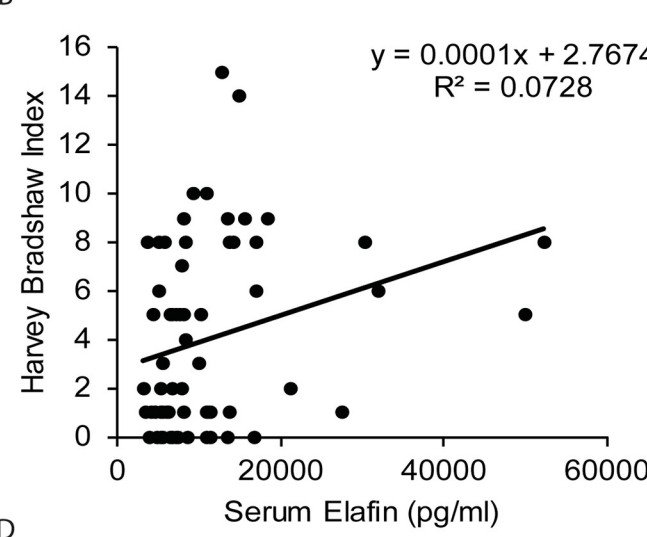

C

D

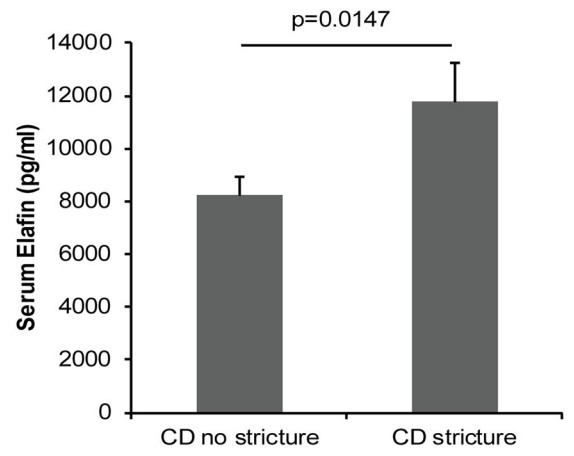

E

|  | no strictue | stricture | Total |
|---|---|---|---|
| Elafin >8000pg/ml | 15 | 13 | 28 |
| Elafin <8000pg/ml | 30 | 7 | 37 |
| Total | 45 | 20 | 65 |
|  | estimated value | 95% CI lower | 95% CI upper |
| prevalence | 0.31 | 0.2 | 0.44 |
| sensitivity | 0.65 | 0.41 | 0.84 |
| specificity | 0.67 | 0.51 | 0.79 |
|  |  |  |  |
| PPV | 0.46 | 0.28 | 0.66 |
| NPV | 0.81 | 0.64 | 0.91 |
|  |  |  |  |
| relative risk | 2.45 | 1.13 | 5.34 |
| SIGNIFICANCE | p=0.0235 |  | significant |

**Fig 1. Circulating elafin levels are increased in IBD patients.** (A) Serum elafin levels of 50 normal, 23 UC, and 28 CD patients in cohort 1 and 20 normal, 57 UC, and 67 CD in cohort 2. Multiple group comparisons were done by one-way ANOVA. (B) Scatter plot shows the moderate correlation between serum elafin levels and clinical disease activity (HBI) in 68 CD patients. (C) Scatter plot shows no association between serum elafin levels and endoscopic disease activity (SES-CD) in 68 CD patients. (D) The stricturing CD patients (n = 20) had significantly higher serum elafin levels than non-stricturing CD patients (n = 45) in a combined dataset. Two-group comparison was done by Student's t-test. (E) Prevalence, sensitivity, specificity, positive predictive value, negative predictive value, and relative risk of elafin test for indicating intestinal stricture in CD patients.

indicating stricture in CD patients (area under curve/AUC = 0.657 using elafin alone) (Fig 2A). We utilized machine learning to develop an algorithm for indicating the presence of intestinal strictures in CD patients (Fig 2B). The optimized trained model using serum elafin levels and commonly available clinical data together is much more accurate than those using either elafin or clinical data alone (AUC = 0.917 using combined data; 0.742 using clinical data alone) (Fig 2C). Therefore, a combination of high serum elafin level and other characteristics are strongly associated with the presence of stricture in CD patients.

## Serum elafin levels are not correlated with endoscopic disease activity in UC patients

High serum elafin levels had a weak positive correlation with increased Partial Mayo Score (PMS) in UC patients, as indicated by low $R^2$ value (Fig 2D). Serum elafin levels had no association with the Mayo Endoscopic subscore in the same set of UC patients (Fig 2E). There was no association found between elafin levels, disease location (E1-3), use of medication, age, BMI, and duration of disease at the time of blood collection (S2 Table, panel A, D-F).

## Colonic elafin mRNA and protein expression were low in stricturing CD patients

Baseline characteristics of the colonic tissue cohort are shown in S3 Table, panel A [25]. Consistent with a previous study [19], UC patients had significantly higher colonic elafin mRNA and protein expression than control non-IBD patients (Fig 3A and 3B). CD patients with stricture had significantly lower colonic elafin mRNA and protein expression than those without stricture (Fig 3C and 3D).

Colonic elafin protein expression in the control patients was weak (Fig 3E). Elafin immunoreactivity was found in the colonic mucosa and lamina propria of UC patients (Fig 3E). Consistent with another study [20], colonic elafin protein expression was low in CD patients with and without stricture (Fig 3E).

When the entire CD patient cohort was divided into tertiles of colonic elafin mRNA and protein expression, the low tertile tended to have a higher incidence of intestinal stricture than the middle and high tertiles (Fig 4A and 4B). This evidence suggests that stricturing CD patients have very low colonic elafin expression. Similarly, we found that colonic elafin mRNA and protein expression have a modestly negative correlation with colonic fibrogenic factors (COL1A2, VIM, and TGF-b1) mRNA expression in CD patients (Fig 4C and 4D). Therefore, stricturing CD patients with increased colonic fibrogenic factor expression have low colonic elafin expression.

Colonic elafin mRNA and protein expression were not associated with clinical disease activity in UC and CD patients (S3A–S3D Fig). Colonic elafin mRNA expression had a modest negative correlation with histology score of the colonic tissues in UC and CD patients (S3E and S3F Fig). The colonic elafin mRNA expression or the presence of intestinal strictures had no association with current use of anti-TNF medication, current use of steroid or

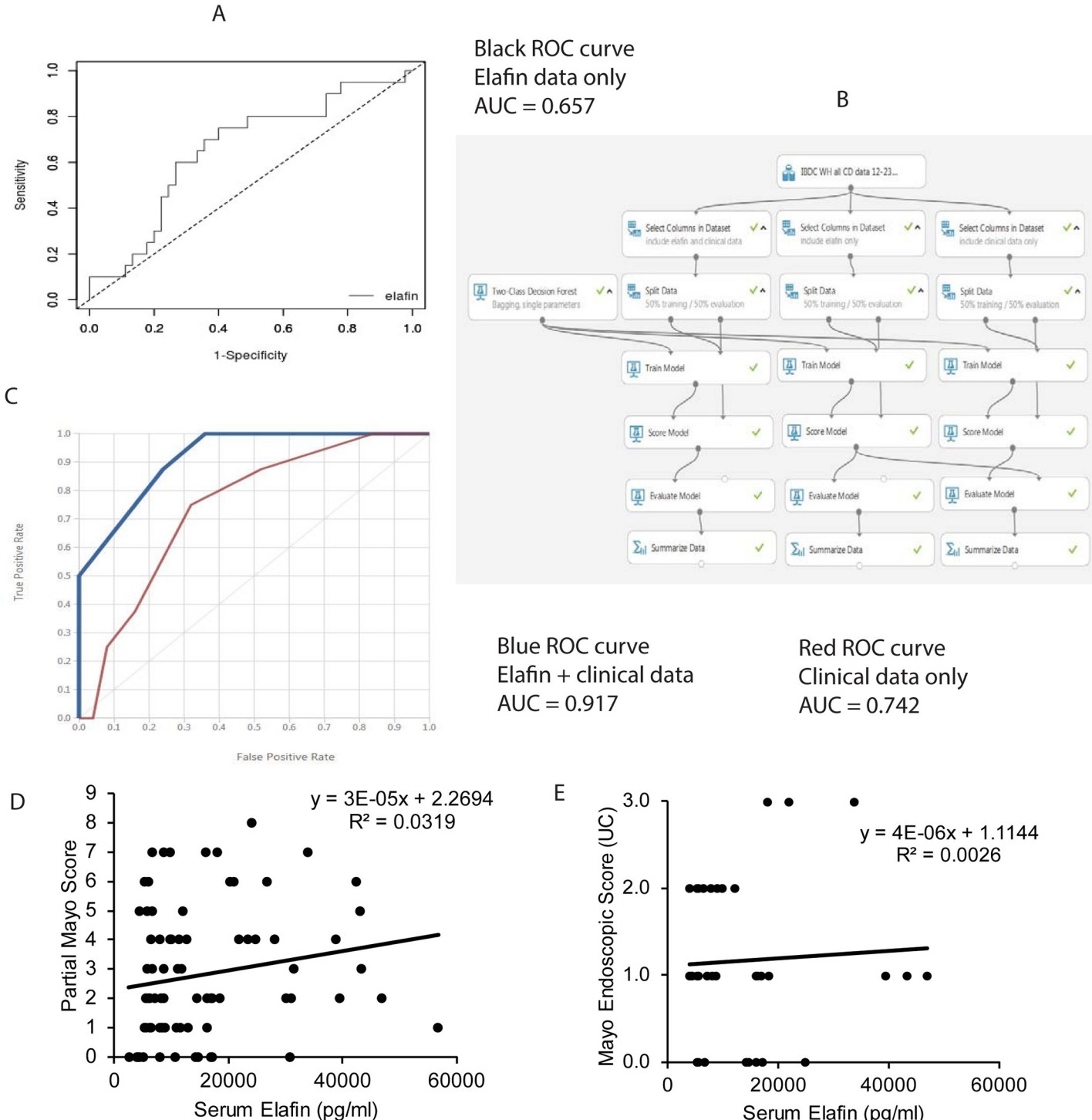

**Fig 2. A combination of serum elafin levels and clinical data indicates the presence of stricture accurately.** (A) ROC curves and AUC values show the accuracy of using elafin data alone for intestinal stricture identification among 67 CD patients (cohorts 1 and 2). The analysis was performed by easyROC web-tool. Cutoff elafin level is 8000pg/ml. (B) The flowchart of Microsoft Azure machine learning algorithms for indicating intestinal strictures in CD patients. (C) ROC curves and AUC values show the accuracy of using clinical data with or without elafin data for intestinal stricture identification among CD patients. (D) Scatter plot shows the positive correlation between clinical disease activity (Partial Mayo Score) and serum elafin levels in 84 UC patients (cohorts 1 and 2). (E) Scatter plot shows no correlation between serum elafin levels and endoscopic disease activity in 36 UC patients (cohorts 1 and 2).

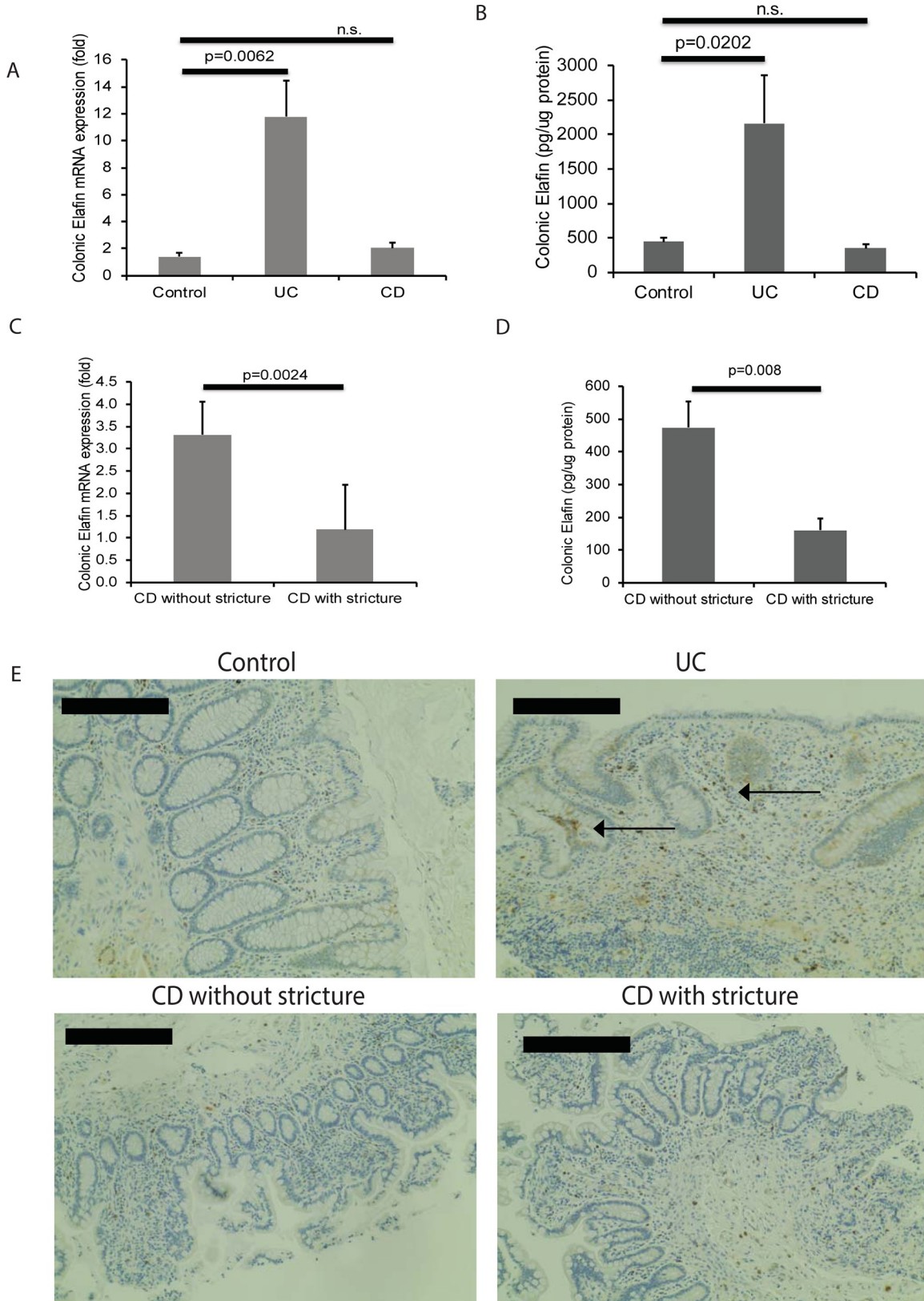

Elafin immunohistochemistry human colon 100X

**Fig 3. Colonic elafin mRNA and protein expression are reduced in stricturing CD patients.** (A) Colonic elafin mRNA expression in 40 non-IBD control, 52 UC, and 52 CD patients (Cedars-Sinai cohort). (B) Colonic elafin protein expression in IBD patients. Multiple group comparisons between control, UC, and CD patients were done by one-way ANOVA. (C-D) Colonic elafin mRNA and protein expression in 15 stricturing CD and 28 non-stricturing CD patients. Two-group comparison between CD with stricture and CD without stricture was done by Student's t-test. (E) Immunohistochemistry of elafin in human colonic tissues. Arrows show the elafin protein in mucosal epithelial layers and lamina propria in UC patients. Six patients per group.

6-mercaptopurine, gender, BMI, age at biopsy collection, or duration of diseases in both UC and CD patients (S3 Table, panel B-C).

## Mesenteric fat adipocytes are a source of circulating elafin in stricturing CD patients

Since stricturing CD patients had high serum elafin levels and low colonic elafin expression, we continued to discover the source of elafin. Stricturing CD patients have a higher visceral to subcutaneous fat area ratio than non-IBD patients [34]. Stricturing CD patients also have a higher visceral fat/total fat mass ratio than non-stricturing CD patients [35]. Mesenteric fat may be a potential source of elafin in circulation.

Sticturing CD patients had significantly higher mesenteric fat elafin mRNA expression than control and non-stricturing CD patients (Fig 5A). Immunohistochemistry indicates that elafin protein expression in mesenteric fat of stricturing CD patients was much higher than those in non-IBD, UC, and non-stricturing CD patients (Fig 5B, left side). At high magnification (400X), the elafin-positive signal is located around adipocytes of stricturing CD patients (Fig 5B, right side).

Interestingly, our patient-matched biopsy collection indicates that mesenteric fat elafin mRNA expression is positively correlated with the mRNA expression of colonic fibrogenic factors (COL1A2, ACTA2, VIM) and negatively correlated with the colonic elafin protein expression in CD patients (Fig 5C and 5D). Therefore, increased elafin expression in mesenteric fat is associated with low elafin expression and intestinal strictures in CD patients.

To identify the link between intestinal strictures and adipose-derived elafin expression, we exposed primary human differentiated mesenteric fat adipocytes to human serum exosomes and determined their elafin secretion. Circulating exosomes mediate long-distance communication between organs and affect disease activity in IBD [36]. When the adipocytes from non-IBD, CD, and UC patients were exposed to normal serum exosomes, their elafin secretion was similar (S5A and S5B Fig). Serum exosomes from stricturing CD patients, but not non-stricturing CD patients, significantly increased elafin secretion of the adipocytes from CD patients (S5A Fig), while serum exosomes from UC patients did not affect elafin secretion of the adipocytes from UC patients (S5B Fig). Therefore, differentiated mesenteric fat adipocytes are a source of elafin in the stricturing CD patients.

## Elafin induces fibrogenesis in human colonic fibroblasts

To determine whether the circulating elafin regulates fibrogenesis, we treated the human colonic CCD-18Co fibroblasts with human sera from normal, stricturing CD, and non-stricturing CD patients (Fig 6A). Sera from stricturing CD patients, but neither from healthy control nor non-stricturing CD patients, significantly increased collagen and ACTA2, but not TGF-b1, mRNA expression in the CCD-18Co fibroblasts (Fig 6A). Exposure to sera from high elafin CD patients also significantly increased COL1A2, but not ACTA2 and TGF-b1, mRNA expression in CCD-18Co fibroblasts (Fig 6B). Neutralization of elafin with anti-elafin antibody

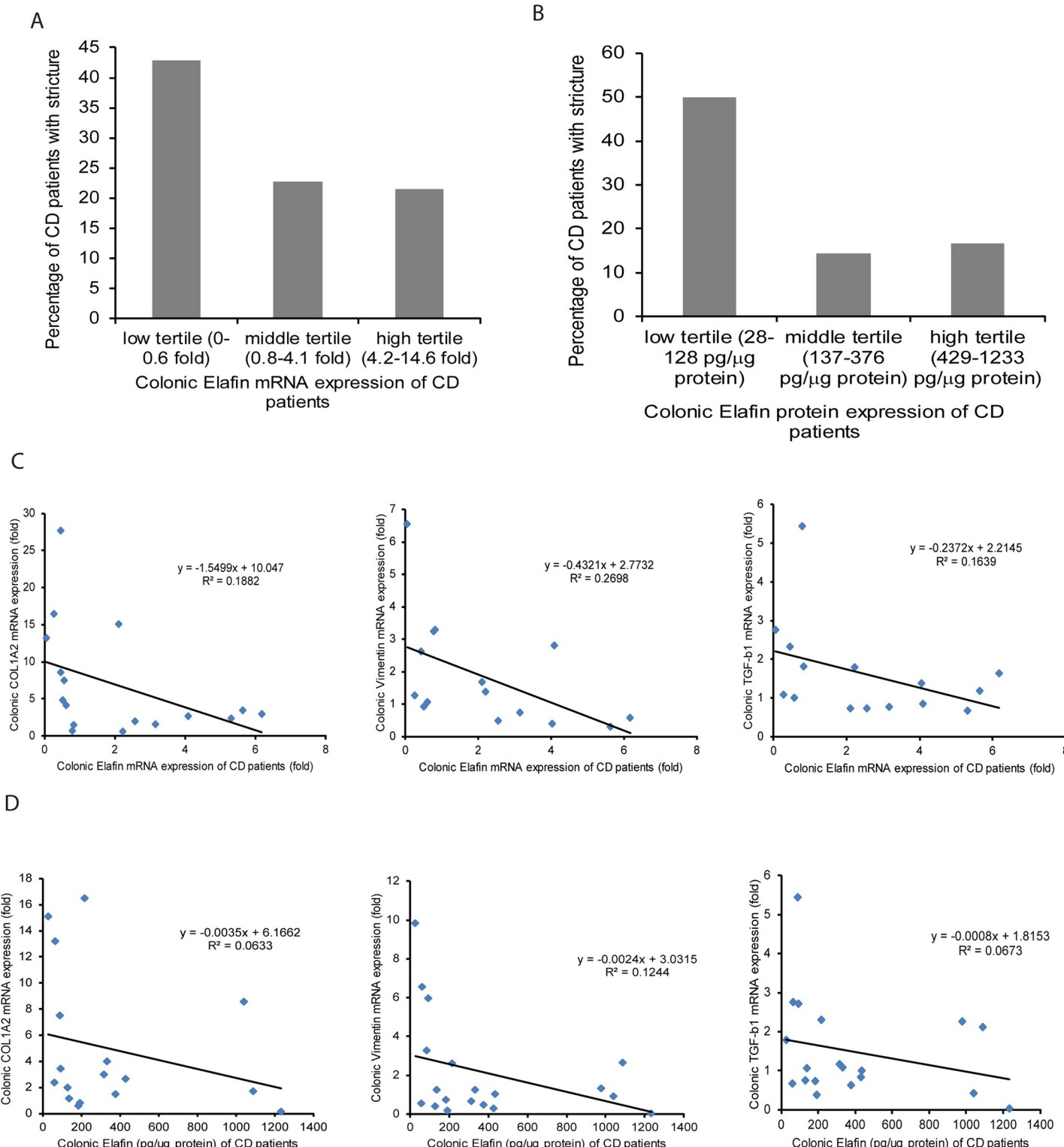

**Fig 4. Colonic elafin mRNA and protein expression are negatively correlated with colonic fibrogenic gene mRNA expression in CD patients.** (A-B) Percentage of intestinal stricture in CD patients assorted by colonic elafin mRNA and protein expression. Low elafin expression groups had a higher percentage of strictures than high elafin expression groups. (C) Scatter plots show the negative correlations between colonic elafin mRNA and colonic collagen (COL1A2), vimentin (VIM), and TGF-b1 mRNA expression in 20 CD patients (Cedars-Sinai cohort). (D) Scatter plots show the negative correlations between colonic elafin protein expression and colonic collagen (COL1A2), vimentin (VIM), and TGF-b1 mRNA expression in 20 CD patients.

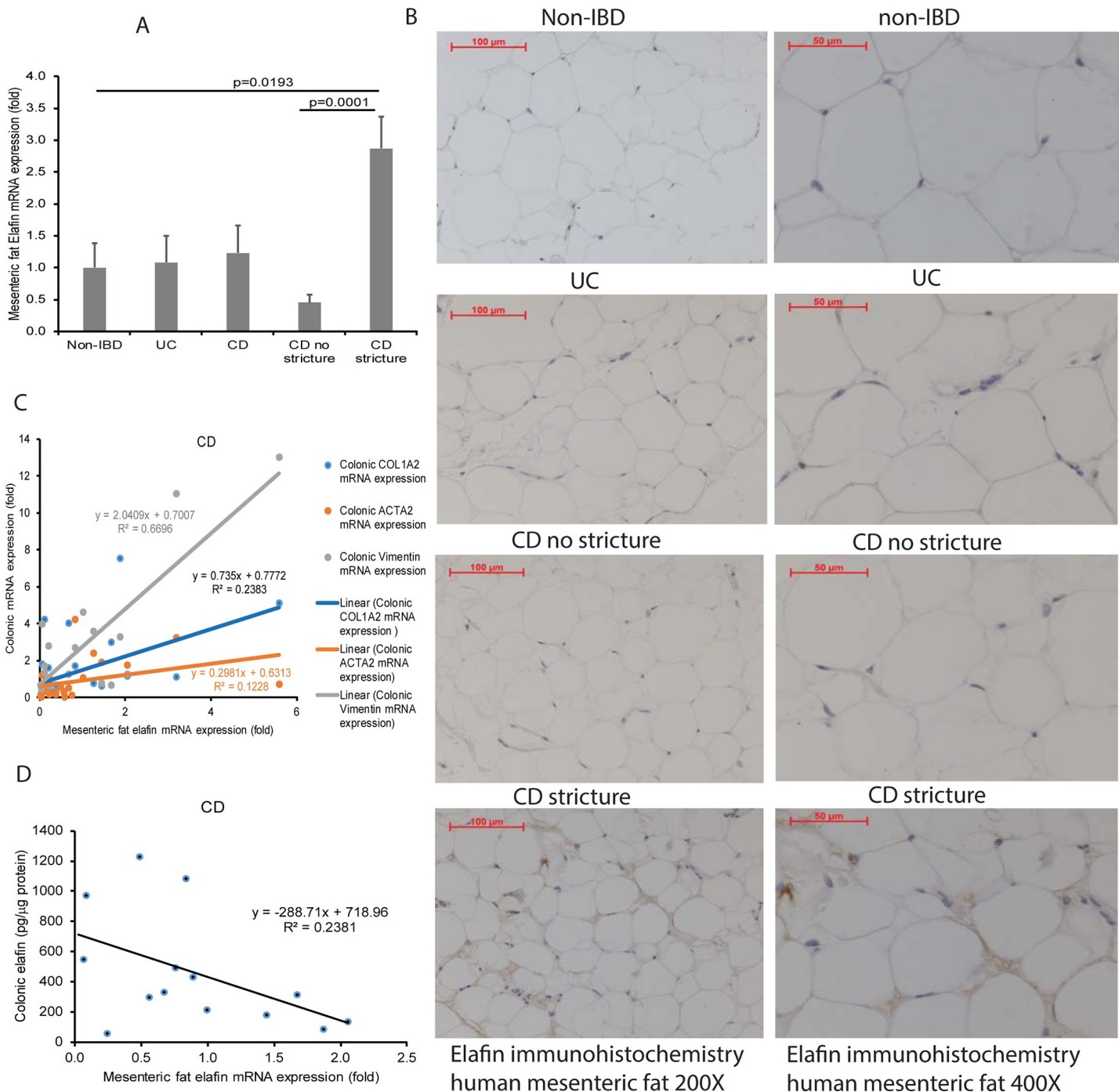

**Fig 5. Mesenteric fat in stricturing CD patients expresses elafin.** (A) Mesenteric fat elafin mRNA expression in 36 non-IBD, 31 UC, 37 CD, 11 non-stricturing CD, and 11 stricturing CD patients (Cedars-Sinai cohort). Multiple comparisons between control, UC, and CD patients were done by one-way ANOVA. Two-group comparison between CD with stricture and CD without stricture was done by Student's t-test. (B) Immunohistochemistry of elafin in human mesenteric fat tissues at 200X and 400X magnifications. Elafin (as shown by brown color) protein expression was strong in mesenteric fat adipocytes in stricturing CD patients. Four patients per group. (C) Scatter plot shows the positive correlation between mesenteric fat elafin mRNA expression and colonic fibrogenic gene mRNA expression in 32 CD patients. (D) Scatter plot shows the negative correlation between mesenteric fat elafin mRNA expression and colonic elafin protein expression in 30 CD patients.

partially reduced the increased collagen mRNA expression in fibroblasts exposed to sera from stricturing CD patients (Fig 6C).

Elafin significantly increased pro-collagen 1A1 protein expression in CCD-18Co fibroblasts (Fig 6D) and increased collagen (COL3A1 and COL1A2), ACTA2, and TGF-b1 mRNA expression in primary human intestinal fibroblasts from CD patients (Fig 6E), suggesting that elafin mediates direct pro-fibrogenic effects on human intestinal fibroblasts.

## Discussion

This report indicates that circulating elafin is associated with intestinal strictures in CD patients. Some of the non-stricturing CD patients also had high circulating elafin levels, leading to moderate accuracy when only elafin was used in identifying stricturing CD patients. Elafin alone is insufficient to indicate intestinal strictures accurately due to the complexity of the patients' many clinical characteristics not taken into consideration. Machine learning improved the accuracy of identifying the presence of intestinal fibrosis among CD patients. Machine learning, a branch of artificial intelligence, is increasingly important for IBD biomarker discovery and disease activity prediction [37, 38]. We have included the Tune Model Hyperparameters module during the tuning and cross-validation step, so the current algorithm has the highest accuracy. The ensemble model of a decision forest worked by voting on the most popular output class of multiple decision trees. Bagging also reduced the chance of overfitting complex models. The resulting algorithm should have high robustness and generalizability. Our predictive experiment is now available on Microsoft Azure AI Gallery (https://gallery.cortanaintelligence.com/Experiment/Use-elafin-and-clinical-data-for-indicating-stricture-Predictive-Exp) for test-run on Microsoft Azure Machine Learning Studio (Classic).

Two expert panels had attempted to establish consensus endpoints and criteria for diagnosis and response to therapy in stricturing CD [39, 40]. Diagnostic approaches for intestinal strictures are based on radiological and endoscopic assessment, which are inherently inconvenient and expensive. The current imaging assessments, including CT and MRE, are unable to differentiate inflammatory versus fibrotic strictures, while some strictures are inaccessible to endoscopy [6]. We are uncertain whether elafin expression is different between inflammatory strictures, fibrotic strictures, and mixed strictures. However, elafin, as a minimally invasive circulating biomarker, may be suitable for identifying high-risk stricturing CD patients for further evaluation. We do not currently have evidence suggesting whether elafin can predict future development of intestinal strictures.

Circulating elafin has moderate sensitivity and specificity in indicating clinical disease activity in CD and UC patients (S1C, S1D, S2A and S2B Figs). The accuracies of elafin in indicating clinical disease activities (AUC = 0.716 in CD and AUC = 0.723 in UC) were similar to the accuracy of CRP (AUC = 0.63 in CD and AUC = 0.70 in UC) (S1E and S2C Figs) [10]. Our biobank is continuing to collect samples and monitor the disease activity of IBD patients. We are optimistic that the associations between elafin and other IBD-related outcomes will be discovered in the future.

To understand the significance of reduced colonic elafin expression in stricturing CD patients, we determined the gene signature in intestinal fibrosis using whole-transcriptome RNA sequencing (RNA-seq). High collagen COL1A2 mRNA expression in the stricturing CD colonic tissue samples indicated fibrosis (S4A Fig). Heat map indicated that the colonic gene expression patterns of stricturing CD and non-stricturing CD patients were different (S4B Fig). Intestinal stricture affected the expression of ~800 genes in CD patients, indicating the specific intestinal host responses to strictures (S4B Fig). Notably, stricturing CD patients consistently had increased fibrosis-related genes such as collagen (COL1A2) and fibronectin (FN1), suggesting that the fibrotic intestinal tissues were occupied by extracellular matrix (S4C Fig, upper panel).

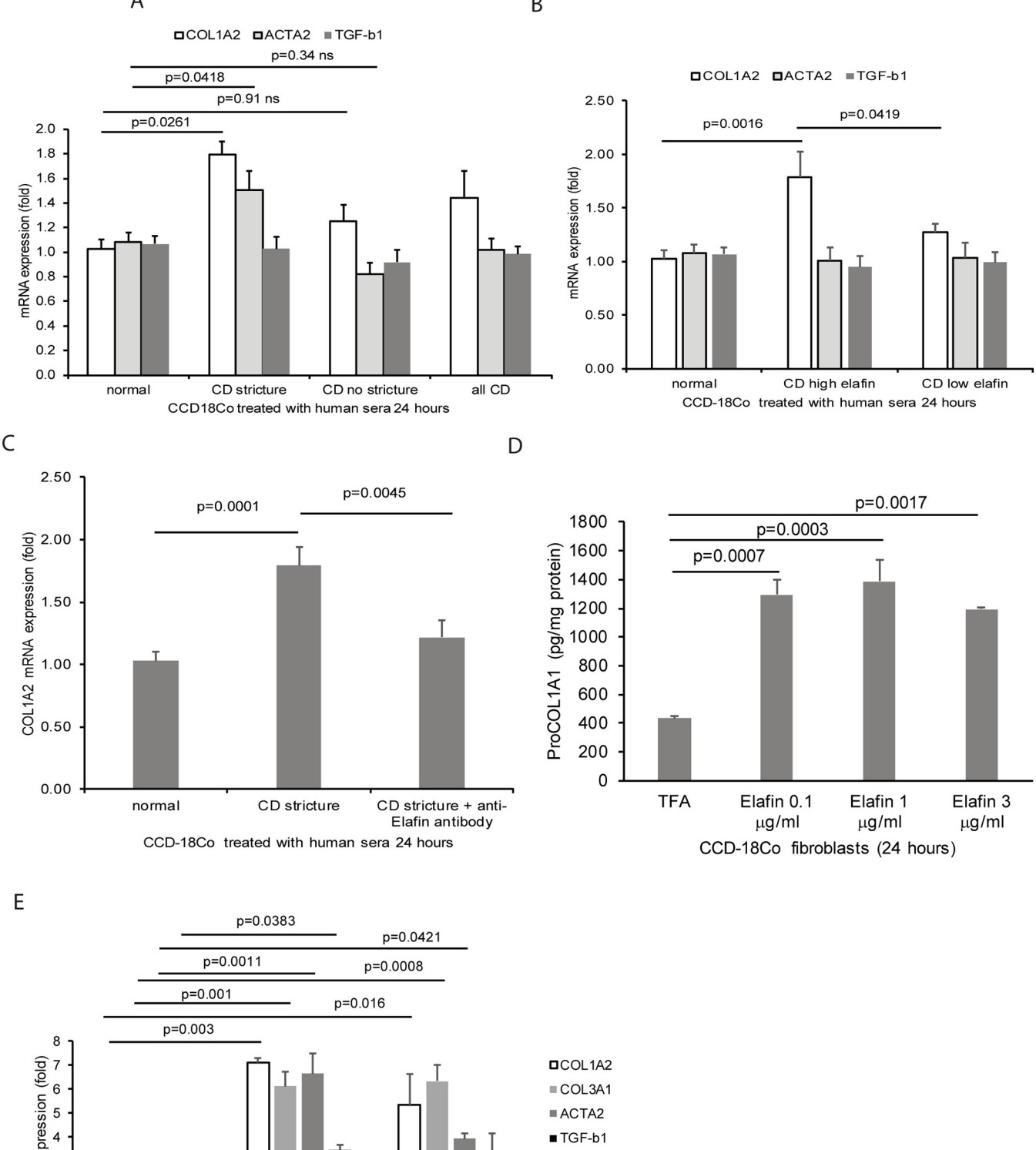

**Fig 6. Elafin promotes fibrogenesis in human colonic fibroblasts.** (A) The human colonic CCD-18Co fibroblasts were incubated with 100μl/ml (10%) of human sera from normal, stricturing CD, and non-stricturing CD patients in serum-free DMEM for 24 hours. (B) The human colonic CCD-18Co fibroblasts were incubated with 100μl/ml (10%) of human sera from high elafin CD group (>8000pg/ml) and low elafin CD group (<8000pg/ml) in serum-free DMEM for 24 hours. (C) Serum-starved CCD-18Co fibroblasts were pretreated with 15μg/ml of anti-elafin neutralizing antibody or control antibody for 30 minutes, followed by exposure to (100μl/ml) human sera from normal and stricturing CD patients for 24 hours. Six serum donors per group. (D) Serum-starved CCD-18Co fibroblasts were exposed to elafin (0.1–3 μg/ml) for 24 hours. Pro-COL1A1 protein levels in cell lysates were determined by ELISA. (E) Serum-starved primary human intestinal fibroblasts from CD patients were exposed to elafin for 24 hours. mRNA expression was determined by real-time RT-PCR.

On the other hand, stricturing CD patients had low expression of epithelial cell-related genes such as keratin (KRT), mucin (MUC), and solute carrier SLC superfamily (S4C Fig, lower panel), suggesting impaired epithelial functioning. Colonic expression of antimicrobial peptide genes such as elafin (PI3) and alpha defensin (DEFA5-6) was also consistently low in stricturing CD patients (S4C Fig, lower panel). Based on these findings, reduced colonic elafin mRNA expression is associated with impaired functioning of the colonic mucosa in the stricturing CD patients.

Colonic mucosa of UC patients has increased antimicrobial peptide expression, such as cathelicidin [31] and beta-defensin 2 [41]. This response may be a protective mechanism against the invasion of luminal bacteria [12, 42, 43]. Since neutrophil accumulation is commonly observed in the colonic mucosa of UC patients [44], the contribution of neutrophil-derived elafin may increase colonic elafin expression, which possibly regulates neutrophil elastase activity and tissue damage in UC patients [45]. Intestinal stricture development involves multiple CD-specific factors. Many UC patients have increased elafin expression (Figs 1A, 3A and 3B), but none of them develop intestinal strictures.

Our study supports the association between adipose tissue and stricture development [34, 35]. The increased mesenteric fat elafin production may be an attempt to compensate for the down-regulated colonic elafin expression by raising circulating elafin levels in the stricturing CD patients (Fig 5). We discovered that serum exosomes from stricturing CD patients induced elafin secretion in mesenteric fat adipocytes from CD patients (S5A Fig), but the exosomal elafin-inducing factors are unknown.

A previous study demonstrated that peripheral blood leukocytes from IBD patients had reduced elafin mRNA expression [21, 22]. Similarly, exposure to serum exosomes from stricturing CD, non-stricturing CD, and UC patients significantly reduced elafin mRNA expression in peripheral blood mononuclear cells (PBMCs) from normal subjects (S5C Fig). PBMCs are not a significant source of circulating elafin in IBD patients.

Serum exosomes from stricturing CD patients induced COL1A2 and ACTA2 mRNA expression in CCD-18Co fibroblasts (S5D Fig). The pro-fibrogenic effects of these serum exosomes was not affected by the circulating elafin levels of the stricturing CD donors (S5E Fig). Therefore, serum exosomes are an elafin-independent pro-fibrogenic factor in CD patients.

Approximately 83–99% of circulating miRNAs are carried by serum exosomes [46]. Therefore, exposure of fibroblasts, adipocytes, or PBMCs to serum exosomes from IBD patients may mimic the circulating environment in IBD. Stricturing CD patients had significantly lower serum exosomal miR205-5p expression than non-stricturing CD patients (S5F Fig). Inhibition of miR205-5p induced COL1A2 mRNA expression in CCD-18Co fibroblasts (S5G Fig). The low serum exosomal miR205-5p expression may be associated with the pro-fibrogenic effect of circulating exosomes from stricturing CD patients because miR205-5p is anti-fibrogenic [47]. The current analysis of serum exosomal miRNAs was limited and did not find the correlations between the tested miRNAs and circulating elafin levels. We will perform RNA sequencing and proteomic analysis of serum exosomes in the future.

Circulating elafin is not associated with the endoscopic severity of colitis in CD and UC patients (Figs 1C and 2E) because colonic elafin expression is not strongly associated with mucosal histology scores in CD patients (S3E Fig). The low colonic elafin expression did not affect the mucosal histology scores in CD patients (S3F Fig). However, it is unfeasible to evaluate the influence of endogenous elafin in the development of intestinal fibrosis in mice because they do not have the elafin gene.

## Conclusions

Our study is the first to recognize elafin as a communication signal between mesenteric fat, blood, and intestine during stricture development. Intestinal stricture is associated with increased circulating elafin levels, reduced intestinal elafin expression, and increased mesenteric fat elafin expression. Machine learning integrated the elafin level and clinical data to develop an improved algorithm for indicating the presence of intestinal strictures in CD patients accurately.

Elafin test may be an adjunct to currently available modes of investigation in CD patients in general. Gastroenterologists need to assess clinical disease activity (HBI) and have required clinical data ready as in current clinical practice. If there is a suspicion of the presence of intestinal strictures in the CD patients, we suggest including elafin tests in regular blood tests along with CRP and ESR during clinical visits of CD. With the required data, the machine-learning algorithm calculates score probability instantly. If the score probability is >0.5, further diagnosis of intestinal strictures, such as endoscopy or imaging, is recommended.

## Supporting information

**S1 Table. Baseline characteristics of serum samples.**
(PDF)

**S2 Table. Disease locations, medications, age, BMI, and duration of disease of IBD patients (Serum sample cohorts).**
(PDF)

**S3 Table. Baseline characteristics of colonic and mesenteric fat samples.**
(PDF)

**S1 Fig. Circulating elafin is moderately accurate in indicating clinical disease activity in CD patients.** (A) The stricturing CD patients had higher serum elafin levels than non-stricturing CD patients in two separate datasets from cohorts 1 and 2. The differences were statistically insignificant. Two-group comparison was done by Student's t-test. (B) Serum elafin levels in 18 CD patients with intestinal fistulas versus 67 CD patients without intestinal fistulas in a combined dataset. The difference was statistically insignificant. (C-D) Prevalence, sensitivity, specificity, positive predictive value, negative predictive value, and odds ratio values of elafin test in indicating (C) CD clinical remission and (D) moderate or severe CD clinical activity. (E) ROC curve with AUC value demonstrates the moderate accuracy of using the elafin test for indicating CD clinical disease activity. Optimal cutoff point is 8000pg/ml.
(PDF)

**S2 Fig. Circulating elafin is moderately accurate in indicating clinical disease activity in UC patients.** (A-B) Prevalence, sensitivity, specificity, positive predictive value, negative predictive value, and odds ratio values of elafin test in indicating (A) UC clinical remission and (B) moderate or severe UC clinical activity. (C) ROC curve with AUC value demonstrates the moderate accuracy of using the elafin test for indicating UC clinical disease activity. Optimal

cutoff point is 18000pg/ml.
(PDF)

**S3 Fig. Colonic elafin mRNA expression is negatively correlated with colonic injury in CD and UC patients.** (A-B) Scatter plots show no significant correlation between clinical disease activity and colonic elafin mRNA expression in UC and CD patients. (C-D) Scatter plots show no significant correlation between clinical disease activity and colonic elafin protein expression in 30 UC and 27 CD patients. Simple Clinical Colitis Activity Score for UC patients. Harvey Bradshaw Index for CD patients. (E-F) Scatter plots show the weak negative correlation between colonic histology score and colonic elafin mRNA expression in 30 UC and 27 CD patients. The analysis included 26 UC patients and 29 CD patients.
(PDF)

**S4 Fig. Colonic gene signatures of stricturing CD and non-stricturing CD patients are different.** (A) Colonic COL1A2 and elafin mRNA expression were determined by real-time RT-PCR and four samples were selected for RNA sequencing. The colonic tissues from stricturing CD patients had high collagen and low elafin mRNA expression. (B) Heat-map of increased (green) and decreased (red) gene expression in the colonic tissues of 2 stricturing CD patients versus 2 non-stricturing CD patients. The RNA-Seq was performed by Omega Biosciences. (C) A list of overexpressed and underexpressed genes in the colonic tissues of CDS patients, compared to CDNS patients. 2 CD patients (HBI = 2) per group. >20-fold increased and >9-fold decreased genes in log2(fold change) were shown.
(PDF)

**S5 Fig. Serum exosomes from stricturing CD patients stimulate elafin secretion in mesenteric fat adipocytes from CD patients.** (A) Serum-starved primary human mesenteric fat adipocytes were exposed to 100μg/ml serum exosomes from normal, stricturing CD (CDS), or non-stricturing CD (CDNS) patients for 16 hours, followed by incubation with serum-free DMEM media for 6 hours. (B) Serum-starved primary human mesenteric fat adipocytes were exposed to 100μg/ml serum exosomes from normal or UC patients for 16 hours, followed by incubation with serum-free DMEM media for 6 hours. Conditioned media were collected from elafin ELISA. Each adipocyte group consisted of 5 patients. (C) PBMCs were exposed to 100μg/ml serum exosomes normal, stricturing CD, non-stricturing CD, and UC patients for 24 hours. (D-E) The human intestinal fibroblasts were incubated with 100μg/ml of human serum exosomes in serum-free DMEM for 24 hours. The collagen (COL1A2) mRNA expression was determined by real-time RT-PCR. Each serum exosome treatment group consisted of 6 patients per group. Multiple group comparison was done by one-way ANOVA. (E) The human colonic CCD-18Co fibroblasts were incubated with 100μg/ml of human serum exosomes from high elafin CD group (>8000pg/ml) and low elafin CD group (<8000pg/ml) in serum-free DMEM for 24 hours. Each serum exosome treatment group consisted of 6 patients per group. (F) Serum exosomal miRNA expression was determined by real-time RT-PCR. (G) Serum-starved CCD-18Co fibroblasts were treated with miR205-5p power inhibitor for 24 hours. Collagen (COL1A2) and ACTA2 mRNA expression were determined by real-time RT-PCR.
(PDF)

## Acknowledgments

We thank Prof. Charalabos Pothoulakis, MD, for technical and financial assistance to this project. The report is associated with a US provisional patent #62/650,981 filed on 3/31/2018.

## Author Contributions

**Conceptualization:** Hon Wai Koon.

**Data curation:** Jiani Wang, Christina Ortiz, Ying Xie, Hon Wai Koon.

**Formal analysis:** Hon Wai Koon.

**Funding acquisition:** Hon Wai Koon.

**Investigation:** Jiani Wang, Christina Ortiz, Lindsey Fontenot, Hon Wai Koon.

**Methodology:** Hon Wai Koon.

**Project administration:** Hon Wai Koon.

**Resources:** Wendy Ho, S. Anjani Mattai, David Q. Shih, Hon Wai Koon.

**Software:** Hon Wai Koon.

**Supervision:** Hon Wai Koon.

**Validation:** Hon Wai Koon.

**Visualization:** Hon Wai Koon.

**Writing – original draft:** Hon Wai Koon.

**Writing – review & editing:** Hon Wai Koon.

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
