## [Decision Letter · Decision Letter 0]

18 Feb 2020

PONE-D-20-00231

High circulating elafin levels are associated with Crohn’s disease-associated intestinal strictures.

PLOS ONE

Dear Dr Koon,

Thank you for submitting your manuscript to PLOS ONE. After careful consideration, we feel that it has merit but does not fully meet PLOS ONE’s publication criteria as it currently stands. Therefore, we invite you to submit a revised version of the manuscript that addresses the points raised during the review process.

In addition to the issues raised by the authors, please comment on this:

I’m really impressed by all the work that has been put into this paper. The material is rather small, but I realize that these patients are scarce.

I’m a bit concerned with the presented data on the multiplex cytokine data. My experience with these assays are that a considerable variation exist especially at low levels. You should include some more on the assay performance (detection level, CV) and some other statics than pure regression (is the observed difference significant). Remember than when you are fishing with so many cytokine, mass significance is an issue. However, you could also delete the cytokine data, I cannot see they add information.

We would appreciate receiving your revised manuscript by Apr 03 2020 11:59PM. To enhance the reproducibility of your results, we recommend that if applicable you deposit your laboratory protocols in protocols.io, where a protocol can be assigned its own identifier (DOI) such that it can be cited independently in the future. For instructions see: http://journals.plos.org/plosone/s/submission-guidelines#loc-laboratory-protocols

We look forward to receiving your revised manuscript.

Kind regards,

Pal Bela Szecsi, M.D. D.M.Sci.

Academic Editor

PLOS ONE

Journal Requirements:

2. In the ethics statement in the Methods and online submission information, please specify the type of informed consent that was obtained from study participants from the Cedars-Sinai Medical Center (for instance, written or verbal, and if verbal, how it was documented and witnessed).

Reviewers' comments:

Reviewer's Responses to Questions

**Comments to the Author**

1. Is the manuscript technically sound, and do the data support the conclusions?

Reviewer #1: Yes

Reviewer #2: Yes

2. Has the statistical analysis been performed appropriately and rigorously? 

Reviewer #1: Yes

Reviewer #2: Yes

3. Have the authors made all data underlying the findings in their manuscript fully available?

Reviewer #1: Yes

Reviewer #2: Yes

4. Is the manuscript presented in an intelligible fashion and written in standard English?

Reviewer #1: Yes

Reviewer #2: Yes

5. Review Comments to the Author

Reviewer #1: Comments to the authors:

1. The title is appropriate and to the point and endeavours to prove the headline statement in the ensuing discussion.

2. In the Introduction section, I suggest a short succinct summary on the mechanisms involved in the fibrogenesis of Crohn’s disease and the link to elafin, clearly establishing the relationship.

3. Also in the Introduction on page 5 line 47 the following sentence should be amended to include the and…but the mechanism of this association is unknown AND has not yet been identified.

4. In the Materials and Methods section it must be made clear that there are 2 separate study groups (a) the serum sample group where only blood was taken and the (b) surgical group where colon and mesenteric fat is sampled. Also, that no blood was collected for the purposes of the study from (b). In the Inclusion and Exclusion criteria it should include whether active IBD at time of recruitment was a requirement for both the two groups mentioned above, especially for (a), or was the IBD quiescent at the time.

5. Further in the Materials and Methods section on page 6 line 47 the following sentence should be reviewed and amended: The healthy subjects did not have concurrent cancer, infection, obesity….. The control group in the surgical group (b) was patients with cancer requiring surgery! Maybe Inclusion and Exclusion criteria for the 2 study groups should be separated.

6. In the Results section the 2 study groups are discussed interchangeably creating the impression that all patients had all procedures. The importance here is that in the serum sample study according to supplementary Table 1 the Partial Mayo score for cohorts 1 and 2 were 3.9 and 4.2 while in the surgical group the colitis score was 6.8. Similarly, the HBI in CD was for the serum sample study was 3.6 and 4.2 for non-stricturing and stricturing disease respectively, while it was 7.4 and 5.0 in the surgical arm. This discrepancy should be explained and the extrapolation justified.

7. Can the authors please explain the discrepancy in the numbers ion Fig1D. The total number of CD patients in the serum study was 95. Where are the rest of the patients as results of only 65 are shown.

8. Finally, in the Results section, the mean age of the control group in surgical arm of the study was 20 years older than the IBD cohorts. Although the results suggest that this should not be of concern please corroborate this a reference as such.

9. The Discussion is generally well written and motivated, but I find some of the conclusions made are due to overinterpretation. From the data, serum elafin will not be a great discriminant for structuring CD as a stand-alone. Prospective testing will also increase the cost as the authors stated that elafin, at this point, is unable to predict future structuring. It should be stated that this may be an adjunct to currently available modes of investigation in CD in general.

10. Finally, the authors must give a schema of the clinical use of elafin in current clinical practice as a conclusion summary.

Thank you,

Sincerely

Erns Fredericks

Reviewer #2: Several issues to be raised:

What is the characteristic of stricturing CD patients, inflammatory/fibrotic or both?Please explain

In cohort 2; samples were taken from the bio bank and is this influence the level/expression of elafin?

Any correlation between elafin expression and endoscopiC score for CD?

How do you measure the quality of colonic and mesenteric fat samples? what percentage of mucosa is involved?Were tissues inflamed or non-inflamed?

Elafin expression/level is also be influenced by several cytokines such as IL-8, and do you look at cytokines expression too?

A total of 4 samples were used for next Gen whole transciriptome RNA seq- (are they the same location of stricture? extend of stricture?small or large bowel stricture? was it inflammatory or fibrotic stricture?

Does duration of disease and BMI influence the elafin expression' please explain further

In discussion subtypes of stricture either inflammatory/fibrotic or mixed potentially may influence the level/expression of elafin?

In the past elafin is less expressed in IBD and how do you explain why is it relevant in CD as opposed to UC?

6. PLOS authors have the option to publish the peer review history of their article (what does this mean?). If published, this will include your full peer review and any attached files.

Reviewer #1: No

Reviewer #2: Yes: Raja Affendi Raja Ali

---

## [Author Response · Author response to Decision Letter 0]

2 Mar 2020

Responses to editor and reviewers’ comments:

Editor Comments:

Thank you for reviewing this manuscript.

I’m really impressed by all the work that has been put into this paper. The material is rather small, but I realize that these patients are scarce.

Thank you for your appreciation and understanding.

I’m a bit concerned with the presented data on the multiplex cytokine data. My experience with these assays are that a considerable variation exists especially at low levels. You should include some more on the assay performance (detection level, CV) and some other statics than pure regression (is the observed difference significant). Remember than when you are fishing with so many cytokines, mass significance is an issue. However, you could also delete the cytokine data, I cannot see they add information.

We removed all multiplex cytokine data, as suggested.

Journal Requirements:

We have modified the manuscript to meet PLOS ONE’s style requirements.

2. In the ethics statement in the Methods and online submission information, please specify the type of informed consent that was obtained from study participants from the Cedars-Sinai Medical Center (for instance, written or verbal, and if verbal, how it was documented and witnessed).

The informed consents at Cedars-Sinai Medical Center were made in written form. We modified the Materials and Methods section – Human colonic and mesenteric fat samples (page 5).

We have removed or changed the data-not-shown-related sections in the manuscript. All data are now supported by figures or tables. Changes in the text are highlighted in yellow color.

Reviewers' comments:

Reviewer #1: Comments to the authors:

We thank reviewer 1 for constructive comments and suggestions.

1. The title is appropriate and to the point and endeavors to prove the headline statement in the ensuing discussion.

Thank you for your appreciation.

2. In the Introduction section, I suggest a short succinct summary on the mechanisms involved in the fibrogenesis of Crohn’s disease and the link to elafin, clearly establishing the relationship.

We included a short summary of the fibrogenesis of Crohn’s disease in the first paragraph of introduction section (page 3).

Also, we added a new reference about the role of elafin in elastic fiber accumulation in dermal fibroblasts, which provide a clue to the role of elafin in fibrogenesis (page 3).

3. Also in the Introduction on page 5 line 47 the following sentence should be amended to include the and…but the mechanism of this association is unknown AND has not yet been identified.

Thank you for pointing it out. We corrected it as suggested. It should be on page 4.

4. In the Materials and Methods section it must be made clear that there are 2 separate study groups (a) the serum sample group where only blood was taken and the (b) surgical group where colon and mesenteric fat is sampled. Also, that no blood was collected for the purposes of the study from (b). In the Inclusion and Exclusion criteria, it should include whether active IBD at time of recruitment was a requirement for both the two groups mentioned above, especially for (a), or was the IBD quiescent at the time.

We fully agree with this point.

In the revised serum sample inclusion criteria, we stated that both cohorts included patients with a wide range of (from remission to severe) clinical and endoscopic disease activity (page 4).

In the revised colonic/fat cohort inclusion criteria, we stated that these IBD patients typically had severe disease activity or severe strictures that were justified for surgical resection (page 5).

5. Further in the Materials and Methods section on page 6 line 47 the following sentence should be reviewed and amended: The healthy subjects did not have concurrent cancer, infection, obesity….. The control group in the surgical group (b) was patients with cancer requiring surgery! Maybe Inclusion and Exclusion criteria for the 2 study groups should be separated.

We fully agree with this point. We now described the serum and colon/fat cohorts separately (page 4-5).

6. In the Results section the 2 study groups are discussed interchangeably creating the impression that all patients had all procedures. The importance here is that in the serum sample study according to supplementary Table 1 the Partial Mayo score for cohorts 1 and 2 were 3.9 and 4.2 while in the surgical group the colitis score was 6.8. Similarly, the HBI in CD was for the serum sample study was 3.6 and 4.2 for non-stricturing and stricturing disease respectively, while it was 7.4 and 5.0 in the surgical arm. This discrepancy should be explained and the extrapolation justified.

We understand that IBD clinics accepted IBD patients with a wide spectrum of patients from remission to severe. Therefore, the average Partial Mayo Scores for UC patients and Harvey Bradshaw Indices for CD patients tend to be low. 

On the other hand, only IBD patients with severe inflammation or strictures needed to have surgical resections. IBD patients with mild disease activity were not sent to surgery. It is understandable to see the tendency for the surgical arm to have high clinical disease activity, as reflected by high PMS or HBI values. 

We described these in the inclusion criteria (page 4-5).

7. Can the authors please explain the discrepancy in the numbers ion Fig1D. The total number of CD patients in the serum study was 95. Where are the rest of the patients as results of only 65 are shown. In the Supplementary Table 1, the total number of CD patients from both cohorts was 95. 

As we mentioned in the inclusion criteria, the diagnosis included multiple diagnostic approaches such as MRE, CT, or endoscopy. The strictures were defined by prestenotic dilation, luminal narrowing, and increased wall thickness.

However, these assessments were not routinely performed for all CD patients. 30 CD patients did not have complete stricture-specific assessments. By reading the patient history and physician notes, we could not confirm whether these patients had strictures or not. To avoid confusion, we excluded these 30 unconfirmed patients from the sensitivity/specificity/relative risk and machine learning algorithm calculations. 

The original Figure 1D is now moved to Figure 1E.

8. Finally, in the Results section, the mean age of the control group in surgical arm of the study was 20 years older than the IBD cohorts. Although the results suggest that this should not be of concern please corroborate this a reference as such.

Our surgical arm reflects the demographic of IBD and colon cancer patients. The median age at diagnosis for colon cancer is around 70 years (colorectal cancer alliance), while the median ages at diagnosis for UC and CD patients are 35 and 30 years respectively (Crohn’s and Colitis Foundation). 

As shown in S3 Table, mean age of non-IBD control group at collection was 62, UC at 43, stricturing CD at 36, and non-stricturing CD at 45. Most IBD patients had surgery 5-10 years after diagnosis. Our surgical arm reflected the demographic situation of IBD and colon cancer patients.

We have included new calculations and showed that age is not associated with serum and colonic elafin expression (S2 Table D and S3 Table B-C).

9. The Discussion is generally well written and motivated, but I find some of the conclusions made are due to over-interpretation. From the data, serum elafin will not be a great discriminant for structuring CD as a stand-alone. Prospective testing will also increase the cost as the authors stated that elafin, at this point, is unable to predict future structuring. It should be stated that this may be an adjunct to currently available modes of investigation in CD in general.

We added a sentence in the conclusion paragraph: Elafin test may be an adjunct to currently available modes of investigation in CD patients. (page 18)

10. Finally, the authors must give a schema of the clinical use of elafin in current clinical practice as a conclusion summary.

Elafin test may be an adjunct to currently available modes of investigation in CD patients in general. Gastroenterologists need to assess clinical disease activity (HBI) and have required clinical data ready as in current clinical practice. If there is a suspicion of the presence of intestinal strictures in the CD patients, we suggest including elafin tests in their regular blood tests along with CRP and ESR during clinical visits of CD. With the required data, the machine-learning algorithm calculates score probability instantly. If the score probability is >0.5, further diagnosis of intestinal strictures, such as endoscopy or imaging, is recommended. (page 18)

Reviewer #2: 

We thank Reviewer 2 (Dr. Raja Affendi Raja Ali) for constructive comments.

What is the characteristic of stricturing CD patients, inflammatory/fibrotic or both? Please explain

As shown by a recent publication, there is no reliable imaging approach to differentiate stricture types: 

Gut. 2019 Jun;68(6):1115-1126. doi: 10.1136/gutjnl-2018-318081. Epub 2019 Apr 3.

Assessment of Crohn's disease-associated small bowel strictures and fibrosis on cross-sectional imaging: a systematic review.

MRE and CT were routinely used to identify strictures, which could not differentiate inflammatory versus fibrotic strictures. Endoscopy may help differentiate inflammatory versus fibrotic strictures. However, the gastroenterologists are less likely to conduct endoscopy to CD patients with strictures or severe disease activity because endoscopic procedures may carry risk or the endoscope cannot access into the stricture sites. Therefore, the clinical data did not clearly indicate inflammatory strictures versus fibrotic strictures. 

We discussed this point in discussion (page 15-16).

In cohort 2; samples were taken from the bio bank and is this influence the level/expression of elafin?

As mentioned in the Materials and Methods section, all samples were collected prospectively. Cohort 2 samples were stored in the biobank and then we retrieved the aliquots afterward. Therefore, the cohort 2 samples were collected in the same way as cohort 1. As you can notice in Figure 1A and S1 Figure A, there are no significant differences in serum elafin levels between the two cohorts.

Any correlation between elafin expression and endoscopic score for CD?

We added a new figure in Figure 1C to show that there was no association between serum elafin levels and endoscopic disease activity in CD patients (SES-CD).

How do you measure the quality of colonic and mesenteric fat samples? what percentage of mucosa is involved? Were tissues inflamed or non-inflamed?

Cedars-Sinai Medical Center Pathology was responsible for selecting tissues for our research. Although we did not know the selection criteria in detail, we observed the H&E-stained colonic tissues and found that the tissues were intact and of full-thickness that included mucosa, submucosa, muscularis, and serosa. 

We verified the histological structures of mesenteric fat samples with H&E staining and confirmed that they are intact adipose tissues.

The percentage of mucosa varies, depending on the tissue orientation of the sections. All included tissues have at least 10% mucosa. The Cedar-Sinai Medical Center surgeons also cut out the patient-matched mesenteric fat tissues separately.

All colonic tissues are involved regions of disease, which are inflamed or strictured. We did not include uninvolved uninflamed regions of IBD samples in this study. We mentioned this in the inclusion criteria (page 5).

Elafin expression/level is also be influenced by several cytokines such as IL-8, and do you look at cytokines expression too?

As shown in the original submission, we noticed the positive correlation between serum elafin and specific cytokines, such as IFN� and IL-5 in UC and CD patients. We also know that serum levels are not associated with IL-8 levels in all IBD patients (data not shown in the original submission). However, we followed the editor’s suggestion by removing the cytokine data from the manuscript. We do not discuss the cytokine data in this revised version.

A total of 4 samples were used for next Gen whole transcriptome RNA seq- (are they the same location of stricture? extend of stricture? small or large bowel stricture? was it inflammatory or fibrotic stricture?

The large bowel, colonic total RNA samples from two stricturing CD and two non-stricturing CD patients (Cedars-Sinai Medical Center) were used for next-generation whole-transcriptome RNA sequencing. We included additional mRNA expression data for these patients in S4 Figure A. High collagen COL1A2 mRNA expression in the stricturing CD colonic tissue samples indicated fibrosis.

However, there is no clinical information available for indicating whether they are inflammatory or fibrotic strictures.

Does duration of disease and BMI influence the elafin expression' please explain further

For CD patients, there is no association between elafin levels, age (A1-3), disease location (L1-4), use of medication, and body mass index (BMI) at the time of blood collection (S2 Table, panel A-E). However, stricturing CD patients have significantly longer durations of disease than non-stricturing CD patients do (S2 Table, panel F). Among stricturing CD patients, the high serum elafin group also had a significantly longer duration of disease than the low serum elafin group (S2 Table, panel F). 

For UC patients, there is no association between elafin levels, disease location (E1-3), use of medication, age, BMI, and duration of disease at the time of blood collection (S2 Table 2, panel A, C-E). 

We addressed these in S2 Table and revised the results section.

In discussion subtypes of stricture either inflammatory/fibrotic or mixed potentially may influence the level/expression of elafin?

The current imaging assessments, including CT and MRE, are unable to differentiate inflammatory versus fibrotic strictures, while some strictures are inaccessible to endoscopy. We do not know whether elafin expression is different between inflammatory strictures, fibrotic strictures, and mixed strictures. However, elafin, as a minimally invasive circulating biomarker, may be suitable for identifying high-risk stricturing CD patients for further evaluation. 

In the past elafin is less expressed in IBD and how do you explain why is it relevant in CD as opposed to UC?

As shown by our data in Figure 1 and 2, serum elafin levels had weak positive correlations with clinical disease activity (Partial Mayo Score and Harvey Bradshaw Index), but not endoscopic disease activity (Mayo Endoscopic Subscore and Simple Endoscopic Index for CD), in IBD patients. Serum elafin levels are useful to identify CD patients with a high risk of strictures, but elafin is not useful to indicate general disease activity.

Similar to two elafin studies (Zhang et al., PMID: 29084078) and (Schmid et al., PMID 17200145), colonic elafin mRNA expression in CD patients is low (Figure 3). However, we found that stricturing CD has lower colonic elafin expression than non-stricturing CD (Figure 3).

Schmid et al. showed that inflamed UC had stronger elafin colonic expression than non-inflamed UC or controls. We also found that colonic tissues from active UC patients had increased elafin mRNA and protein expression, but we do not understand why Zhang’s study showed reduced elafin expression in UC patients.

Nevertheless, our results are at least partly consistent with the findings of other groups. 

I agree that elafin plays different roles in CD and UC. Due to the limit of the scope of this manuscript, we cannot know all aspects of elafin in IBD. However, we managed to discover the unique role of elafin in CD strictures.

---

## [Decision Letter · Decision Letter 1]

1 Apr 2020

High circulating elafin levels are associated with Crohn’s disease-associated intestinal strictures.

PONE-D-20-00231R1

Dear Dr. Koon,

We are pleased to inform you that your manuscript has been judged scientifically suitable for publication and will be formally accepted for publication once it complies with all outstanding technical requirements.

With kind regards,

Pal Bela Szecsi, M.D. D.M.Sci.

Academic Editor

PLOS ONE

Additional Editor Comments (optional):

Reviewers' comments:

Reviewer's Responses to Questions

**Comments to the Author**

1. If the authors have adequately addressed your comments raised in a previous round of review and you feel that this manuscript is now acceptable for publication, you may indicate that here to bypass the “Comments to the Author” section, enter your conflict of interest statement in the “Confidential to Editor” section, and submit your "Accept" recommendation.

Reviewer #1: All comments have been addressed

Reviewer #3: All comments have been addressed

2. Is the manuscript technically sound, and do the data support the conclusions?

Reviewer #1: Yes

Reviewer #3: Yes

3. Has the statistical analysis been performed appropriately and rigorously? 

Reviewer #1: Yes

Reviewer #3: Yes

4. Have the authors made all data underlying the findings in their manuscript fully available?

Reviewer #1: Yes

Reviewer #3: Yes

5. Is the manuscript presented in an intelligible fashion and written in standard English?

Reviewer #1: Yes

Reviewer #3: (No Response)

6. Review Comments to the Author

Reviewer #1: The authors have addressed all the concerns and answered all the queries raised in the first round of review. I feel comfortable that all requirements have been met and that the revised manuscript can proceed to publications. It is a complicated study with many moving parts, but in the end I feel the authors attempted to answer the questions raised in the aims of the study. Congratulations.

Reviewer #3: (No Response)

7. PLOS authors have the option to publish the peer review history of their article (what does this mean?). If published, this will include your full peer review and any attached files.

Reviewer #1: Yes: Ernst Fredericks

Reviewer #3: Yes: Huahong Wang

---

## [Editor Report · Acceptance letter]

2 Apr 2020

PONE-D-20-00231R1 

High circulating elafin levels are associated with Crohn’s disease-associated intestinal strictures. 

Dear Dr. Koon:

I am pleased to inform you that your manuscript has been deemed suitable for publication in PLOS ONE. Congratulations! Your manuscript is now with our production department. 

With kind regards,

on behalf of

Dr. Pal Bela Szecsi 

Academic Editor

PLOS ONE